# Traceable Features of Static Plantar Pressure Characteristics and Foot Postures in College Students with Hemiplegic Cerebral Palsy

**DOI:** 10.3390/jpm12030394

**Published:** 2022-03-04

**Authors:** Tong-Hsien Chow

**Affiliations:** Department of Leisure Sport and Health Management, St. John’s University, New Taipei 25135, Taiwan; thchow1122@mail.sju.edu.tw

**Keywords:** cerebral palsy (CP), centers of gravity, hemiplegic limbs, plantar pressure distributions (PPDs), pronated low arches, rearfoot valgus

## Abstract

Patients with cerebral palsy (CP) are characterized by disturbances of mobility with postural and foot deformities. Subsequent development of CP may lead to changes in plantar loading. This study examined the characteristics of foot types and relative loads associated with centers of gravity and foot posture in college students with left and right hemiplegic CP, as well as these differences between unaffected and hemiplegic limbs. A cross-sectional study of 45 hemiplegic college students with mild CP and 62 healthy students was conducted. Static plantar pressure was measured with a JC Mat. CP students exhibited low arches, and their plantar pressure distributions (PPDs) were mainly exerted on the left forefoot, as well as on the right forefoot and rearfoot. The weight shifted to the unaffected foot with dual plantar loading regions (forefoot and rearfoot), rather than the hemiplegic foot with a single region (forefoot). PPDs commonly increased at the medial metatarsals of both feet, and hemiplegic CP students presented the increased PPDs on the medial aspect of the hemiplegic foot accompanied by a rearfoot valgus posture pattern. The findings revealed a traceable feature to a possible connection among the pronated low arches, mild centers of gravity, metatarsal syndrome and rearfoot valgus of the hemiplegic limbs in CP patients.

## 1. Introduction

Patients with cerebral palsy (CP) often have a permanent disorder of mobility, muscle tone, and posture development, which limits their function in daily activities [1]. CP is related to the abnormal signal transmission between the brain and the extremities [1], and causes various symptoms, including spasticity, contracture and uncoordinated movement, and this, then, affects movement abilities [2]. The symptoms of CP usually manifest as a variety of dyskinetic effects on the extremities of the body. The influences are often manifested as contractures of the musculoskeletal system, which result in hyperactivity and balance problems [3]. Spasticity of the lower extremities can lead to toe-in gaits and crouch gaits, which can lead to falling and unstable balance while walking [4]. People with CP have difficulty walking due to foot deformities and poor control of their muscles and motor skills [5]. Deformities of the feet are common in people with CP, resulting in unnatural foot landings and abnormal foot pressure distributions, as well as changes in posture and gait [6]. These changes may affect the alignment of the foot and ankle, and could lead to specific podiatric diseases [7]. Although current research has extensively explored functional limitations of gait and body posture in children with CP through motion analysis [8,9], however, few quantitative studies have examined morphological and functional characteristics of the foot in CP patients [6].

Plantar pressure is considered to be an effective indicator of the distribution of plantar loads, abnormal gaits, lower limb alignment, podiatric severity, and rehabilitation condition [10] as well as the ability to confirm movements [11,12]. The parameters of footprints can be used to provide important information about the link between multiple regions of the foot and anatomical function [13], which is beneficial for diagnosing foot pathology, preventing foot deformities, and treating and rehabilitating podiatric diseases. [14,15]. Static measurement of plantar pressure may also help clarify the relationship between plantar loading and foot posture, thus plantar pressure measurement technology is useful for studying the biomechanics of the foot and lower limb in subjects [16]. Plantar pressure measurements clearly reflect changes in foot postural alignment and serve as an ideal tool for investigating toe-in gait in children with CP [17]. Femery et al. reported that children with hemiplegia exhibit marked differences in plantar load distribution between affected and unaffected limbs, especially for the hallux, first metatarsal head and midfoot [18]. Leunkeu et al. delved further, and identified that plantar pressure during walking increased at the medial heel, and decreased at the hallux and lateral heel in hemiplegia, yet, increased the plantar loads on the first metatarsal, the medial and lateral midfoot in diplegia [19]. Galli et al. found that children with CP present antepulsion posture and statically showed lower arched foot [6]. The foot deformity was more common in diplegia than in hemiplegia. The average static plantar pressure in children with diplegic and hemiplegic CP was higher on the forefoot and midfoot regions, while lower on the rearfoot region [6]. In addition to plantar pressure distribution, foot posture index (FPI), heel and knee alignment are considered to be the key factors of affecting foot pressure and the feasible methods to assess variations of the angle and motion behavior of lower extremities and lower limb defects [20,21]. As in the study by Yan et al. [22], they observed that children experienced unstable walking due to a flatter foot and a larger foot axis angle as well as altered foot pressure while walking, compared with non-obese children. Gijon-Nogueron et al.’s large study of pediatric foot posture showed that pronated or flat feet represent the common foot postures in childhood with their corresponding FPI scores [23]. Jiménez-Cebrián et al. [20], further highlighted the relationship among foot type, FPI and foot/lower limb pain.

According to the previous literature, the disorder was often investigated using a force platform to assess postural sway and thereby static balance abilities [24]. Moreover, the research subjects were mostly children who did not distinguish the differences between the left and right side of hemiplegia. Only a few studies have described static foot types, plantar loads, centers of gravity and foot posture in college and university students with CP, and even fewer studies have examined these differences among age-matched male and female students with CP suffering from either left or right hemiplegia. Given that the subsequent development of CP may lead to alternations in foot pressure distributions and foot shapes in CP patients, understanding static foot pressures and foot shape characteristics may be beneficial for school-age students to provide strategies beyond orthotics for improving foot problems and conveniences in life. Therefore, one of the study aims was to investigate the characteristics of the foot arch index (AI), the plantar pressure distributions (PPDs), the balance of the centers of gravity, and the rearfoot postural alignment among college and university students with hemiplegic CP. The other study aim was to determine these differences between hemiplegic and unaffected limbs in hemiplegic CP students. We hypothesized that students with hemiplegic CP exhibit unique characteristics of foot types and plantar loads associated with centers of gravity and foot posture compared to healthy students. Moreover, substantial differences in these research items are observed between left and right hemiplegic CP conditions and between hemiplegic and unaffected limbs in hemiplegic CP students.

## 2. Materials and Methods

### 2.1. Participants

This study involved 107 Taiwanese male and female college students who were divided into two groups: 62 healthy students (the control group) and 45 students diagnosed with hemiplegic cerebral palsy (the CP group). Participants in this study were recruited from St. John’s University, China University of Technology, Taipei City University of Science & Technology, JinWen University of Science & Technology, Aletheia University, Ching Kuo Institute of Management and Health, Kun Shan University, Tungnan University, MingDao university, Taipei University of Marine Technology, Minghsin University of Science and Technology in Taiwan between November 2019 and October 2021.

The inclusion criteria for the CP group (20 males and 25 females) were those diagnosed with mild CP and met the Gross Motor Function Classification System (GMFCS) level I and II based on clinical exams and physician certifications. Level I indicated that patients could unrestrictedly walk. Level II suggested that patients had difficulty walking long distances or with balance. The inclusion criteria were referred to the study by Galli et al. [6]. The criteria for the qualified participants were as follows: (a) being able to walk unrestricted without the support of an Ankle Foot Orthosis (AFO); (b) being able to stand up independently and keep their balance for at least 2 min without assistive devices; (c) being able to stand upright with complete plantar contact with the ground; (d) having not undergone any orthopedic surgery on lower extremity joints in the past; (e) the lower limbs were not injected with botulinum toxin within the previous six months; and (f) having no related problems which may affect balance (e.g., mental deficits, significant visual, hearing and sensory impairments). In addition, participants in the CP group were subdivided into left and right hemiplegic CP groups, which was based on related studies verifying the differences in plantar loads, foot deformity and gait pattern between the left and right hemiplegic CP patients [6,25]. There should be a difference in foot types between the two hemiplegic CP groups. All of them were assessed by a physician, physiotherapist and occupational therapists, and certificated with clinical medical records. There was a 28.6% (45/63) drop-out rate in the recruitment of eligible hemiplegic CP students for the following reasons: (1) having used an AFO or any other orthoses during the past year; (2) physician’s certificate of past fractures or surgeries; (3) having received botulinum toxin within the previous six months; (4) absence rate from the study.

The control group (28 males and 34 females) consisted of healthy college students. The exclusion criteria for this study were history of surgery on the lower extremity (e.g., fractures, trauma, osteotomies, and any other corrective musculoskeletal surgeries over the past six months), with other musculoskeletal disorders not related to cerebral palsy as well as difficulty standing and walking without wearing orthoses or assistive devices. Considering body weight has been shown to affect arch height and foot pressure characteristics in past studies, a strong association was observed between overweight participants and flat feet [26,27]. Consequently, the participants in the present study were required to have a BMI between 18.5 and 22.9. This range has been defined as a normal healthy weight by the World Health Organization (WHO) and Asia Pacific guidelines [28,29]. During the recruitment process for the eligible healthy college students, there was a 27.9% (62/86) drop-out rate based on: (1) abnormal BMI; (2) having professional training in sports disciplines; (3) physician certification for previous fractures or surgeries; (4) absence rate from the study. Detailed demographic information about all participants is shown in Table 1. Before the experiment was conducted, each participant had to sign an informed consent form. All experiments in this study were conducted in compliance with the guidelines and recommendations of the National Taiwan University Research Ethics Committee.

### 2.2. Instruments and Equipment

Analyses of plantar pressure distribution were performed using the foot pressure detector (JC Mat, View Grand International Co Ltd., New Taipei City, Taiwan). The sensing pad of the JC Mat consists of 13,600 sensors on either side of the measuring platform (32 × 17 cm). It offers a delicate plantar pressure image with dots and contains a wide working area for foot pressure sensing. The subject’s arch index (AI), plantar pressure distributions (PPDs), centers of gravity, and bare images of the footprint can all be captured immediately. Through the JC Mat built-in FPDS-Pro software (View Grand International Co Ltd., New Taipei City, Taiwan), both feet can be analyzed and qualified for parameters including AI, PPDs, centers of gravity, toe angles, and footprint characteristics. The repeatability and reproducibility of the device were demonstrated in previous studies [30,31,32,33].

### 2.3. Plantar Pressure Distribution Assessment Processes

To ensure consistency and reliability in the research process, the experiments in this study were conducted between 1 and 4 pm from Monday to Friday of the week. All participants were asked to measure their height, weight, and BMI before the experiment. To obtain accurate data on static footprints, each participant followed these steps:Assist the subject to roll both trouser legs above the knee in order to prevent the trouser legs from restricting the natural stance of the foot;Guide the subject to stand barefoot on the sensing pads with specific marks and the pressure sensing range platform of the JC Mat;Ask the subject to relax, control and balance their centers of gravity in a natural static posture. Stand still with feet shoulder-width apart and distribute body weight evenly on the feet;Ask the subject to stampede easily in place for six to eight steps, then stand still naturally with arms hanging down;Guide the subject to face the experimental instructor standing in front of the subject. Face the instructor directly in the eyes. Keep the body posture steady and balanced until there is no noticeable change in the measurement of foot pressure on the JC Mat.

As participants reached step 5, the JC Mat directly captured and recorded the static foot pressure profiles from footprints.

### 2.4. Plantar Pressure Distribution Measurement

Following capture of the footprint image, three regional and six subregional PPDs were analyzed using the FPDS-Pro software. The software generated the first vertical line from the footprint image. The vertical line extended from the base of the second toe to the heel center, and intersected the most anterior and posterior part of the footprint apart from the toes. Further, the software created four parallel lines which were perpendicular to the first vertical line, and divided the footprint into three equal areas (termed regions A, B, and C) and six subregions (termed subregions 1 to 6). Footprint regions A, B, and C were defined as the forefoot, midfoot, and rearfoot, respectively. Subregions 1 to 6 were divided from the three regions, which were defined as the lateral metatarsal bone (LM), the lateral longitudinal arch (LLA), the lateral heel (LH), the medial metatarsal bone (MM), the medial longitudinal arch (MLA), and the medial heel (LH), respectively. Additionally, the study used a formula for calculating the AI ratio that was developed by Cavanagh and Rodgers [34]. The AI ratio is determined as the ratio of the middle third (midfoot) of the footprint divided by the entire footprint (forefoot, midfoot and rearfoot) excluding the toes, i.e., AI = A/(A + B + C). Based on the study, an AI below 0.21 is considered a high arch, 0.21 to 0.26 is commonly regarded as a normal arch, and above 0.26 is classified as a low arch.

### 2.5. Rearfoot Postural Assessment Processes

In this study, each participant was instructed to stand upright on a fixed platform with their feet at a natural width (approximately 12–15 cm). When standing stably on the platform, a digital camera with a fixed position was used to capture the posterior view of the rearfoot postural alignment image of each participant (an image of at least 754 pixels × 96 pixels) [33]. From the study by Ribeiro et al. [35], the changes in the static rearfoot angle can be calculated as follows:

First, check that the participant’s heel is relaxed and stands on the same horizontal line, then acquire digital photos, identifying the three anatomical points on the lower back of the calf from the digital photo images: (1) the center of the calcaneal tuberosity; (2) the center above the calcaneus; (3) the center of the lower third of the leg. In addition, double lines can be automatically generated from the 3-point connections using the Bio-mech 2019-postural analysis software (Loran Engineering SrL, Castel Maggiore, BO, Italy). The standard straight line (solid line) of the lower extremity was derived from the center of the lower third of the leg to the center above the calcaneus. A second flip angle line (a dotted line) was presented, which connects the center of the calcaneal tuberosity with the center above the calcaneus. A flip angle resulting from the intersection of two straight lines was calculated as the static rearfoot angle. The static rearfoot angle can be classified as a varus foot (<0°), a normal foot (0° to 5°), and a valgus foot (>5°) [36]. As shown in Figure 1, the value of a rearfoot postural alignment was obtained from digital photo images of the representative participant in both the left and right hemiplegic CP groups.

### 2.6. Statistical Analysis

The descriptive statistics were used to compare the age, height, weight, and BMI values of study participants, and results were calculated as mean ± standard deviation (SD). The independent sample *t*-test was employed to compare the AI values, centers of gravity, three regional and six subregional PPDs among groups. Significant statistical differences were defined as *p* < 0.05 and *p* < 0.01. Statistical analyses in the study were conducted using the software program (IBM SPSS Statistics 20, Chicago, IL, USA).

## 3. Results

### 3.1. Arch Index

The bipedal AI values of both feet in static standing of the male and female students with CP were significantly higher than the control group. In the left hemiplegic CP group, the AI values of the left foot were higher among the male and female CP students. Yet, in the right hemiplegic CP group, the AI values of the right foot were also found to be higher in the male and female CP students. The results showed that both male and female students with CP had low-arched feet, and both groups of hemiplegic CP students revealed a low arch pattern in their corresponding hemiplegic limb (Table 2).

### 3.2. Plantar Pressure Distribution of the Three Regions

The PPDs values in this study were expressed as a percentage of relative loading. The static relative loads in the CP group were found to be higher on the left forefoot, and dual regions of the right forefoot and rearfoot than that of the control group, particularly in the male CP students. Yet, the relative loads were mostly distributed over the forefoot region of both feet in the female CP students.

Additionally, in the left hemiplegic condition, the relative loads were predominantly focused on the left forefoot and the right rearfoot. The male students revealed increased relative loads on the dual regions of the forefoot and rearfoot of the unaffected right foot. Apart from this, the female students also showed relatively higher loads distributed on the dual regions of the forefoot and rearfoot of the unaffected right foot, as well as the forefoot of the left foot. With regards to the right hemiplegic state, the relative loads were commonly placed on the forefoot region of both feet, particularly in female students (Table 3).

### 3.3. Balance of the Centers of Gravity

The centers of gravity were presented as a percentage of the gravity. Compared with the control group, the centers of gravity in static standing were found to exhibit a significantly heavier trend on the right foot, particularly in the male CP group.

Nevertheless, in the left hemiplegia, the centers of gravity for the students were heavier for the right foot and lighter for the left foot. Conversely, the centers of gravity in the right hemiplegic CP group were found to be heavier on the left foot while lighter on the right foot. The results indicated that the students in respective hemiplegic CP groups tended to center their gravity on the unaffected limb (Table 4).

### 3.4. Plantar Pressure Distribution of the Six Subregions

The findings of the study revealed that the static relative loads in the CP students were mainly distributed throughout the medial aspects of both feet. When compared with the control group, the CP group’s six subregional relative loads were substantially increased at the medial metatarsals (left foot: 23.86% ± 5.93%; right foot: 25.97% ± 6.23%; *p* < 0.05) of both feet, the medial longitudinal arch (1.07% ± 0.43%; *p* < 0.01) of the left foot and the lateral heel (25.59% ± 5.01%; *p* < 0.05) of the right foot. In addition, the CP group’s relative loads were much lower at the lateral longitudinal arch (15.89% ± 4.44%; *p* < 0.05) and the lateral heel (19.53% ± 6.76%; *p* < 0.01) of the left foot (Figure 2).

In addition, the male CP group’s six subregional relative loads were primarily distributed at the medial metatarsals (left foot: 22.54% ± 7.40%; right foot: 27.57% ± 8.28%; *p* < 0.05) and the lateral heels (left foot: 23.86% ± 4.21%; right foot: 28.05% ± 5.10%; *p* < 0.05) of both feet, and the medial heel (17.58% ± 7.70%; *p* < 0.05) of the right foot (Figure 3).

As for the female CP group, the six subregional relative loads were mostly concentrated at the medial metatarsal (24.70% ±3.60%; *p* < 0.01) of the right foot and the medial longitudinal arches (left foot: 1.04% ± 0.44%; right foot: 0.96% ± 0.48%; *p* < 0.05) of both feet, whereas, the relative loads were decreased at the lateral longitudinal arch (17.71% ± 4.51%; *p* < 0.01) and the lateral heel (16.07% ± 6.45%; *p* < 0.01) of the left foot (Figure 4).

### 3.5. Plantar Pressure Distribution of the Six Subregions in the Left Hemiplegic CP Students

In an analysis of the left hemiplegic CP students, the relative loads of the six sub-regions were mainly centered on the medial metatarsals (left foot: 23.98% ± 7.51%; right foot: 25.23% ± 7.63%; *p* < 0.01) and the medial heels (left foot: 17.52% ± 4.70%; right foot: 15.80% ± 7.57%; *p* < 0.05) of both feet. Additionally, it was observed that it decreased mainly along the lateral longitudinal arch (16.82% ± 4.91%; *p* < 0.01) and the lateral heel (16.08% ± 6.59%; *p* < 0.01) of the left foot (Figure 5).

Further, in the male left hemiplegic CP group, the relative loads were primarily loaded on the medial metatarsals (left foot: 20.93% ± 9.34%; right foot: 26.95% ± 10.80%; *p* < 0.01) of both feet, the medial heel (20.19% ± 8.97%; *p* < 0.01) and the lateral heel (27.08% ± 5.53%; *p* < 0.05) of the right foot (Figure 6). As for the female left hemiplegic CP group, the loads were mainly directed toward the medial metatarsals (left foot: 26.42% ± 4.67%; right foot: 23.85% ± 3.43%; *p* < 0.05) of both feet, the medial longitudinal arch (0.99% ± 0.41%; *p* < 0.01) of the left foot and the lateral heel (25.77% ± 1.34%; *p* < 0.01) of the right foot (Figure 7).

### 3.6. Plantar Pressure Distribution of the Six Subregions in the Right Hemiplegic CP Students

Based on the results of right hemiplegic CP students, the relative loads were discovered to be higher at the medial metatarsal (27.08% ± 3.03%; *p* < 0.01) of the right foot, while lower at the lateral metatarsal (22.12% ± 2.00%; *p* < 0.05) of the left foot (Figure 8) as compared with the control group.

In the male right hemiplegic CP group, the relative loads were mostly at the medial metatarsals (left foot: 24.97% ± 0.59%; right foot: 28.49% ± 1.00%; *p* < 0.05) of both feet and the lateral heel (26.83% ± 0.84%; *p* < 0.05) of the left foot, and relatively decreased at the medial heel (11.25% ± 1.31%; *p* < 0.01) of the left foot (Figure 9). Furthermore, the relative loads of the female right hemiplegic CP group were mainly at the medial metatarsal (25.96% ± 3.65%; *p* < 0.05) of the right foot, the medial heel (16.01% ± 2.03%; *p* < 0.05) and the lateral heel (23.02% ± 0.84%; *p* < 0.05) of the left foot (Figure 10).

### 3.7. Footprint Image Characteristics

The static footprint image of the representative participant in each group is shown in Figure 11. Based on the illustration, the relative loads in the CP group were mainly distributed at the medial metatarsals, the medial longitudinal arch, and the entire heel area of both feet. In addition, the relative loads increased at the medial metatarsals and the medial longitudinal arch of both feet, and the entire right heel in the left hemiplegic CP group, while being mainly distributed at the medial metatarsals of both feet and the left heel in the right hemiplegic CP group.

### 3.8. Rearfoot Postural Alignment

The results of this study indicated that the left hemiplegic CP group’s static rearfoot postural alignment values were higher on the left foot and lower on the right foot, particularly among the males. Conversely, the values in the right hemiplegic CP group were relatively higher on the right foot and lower on the left foot among both males and females.

Based on the results, the hemiplegic foot displayed a rearfoot valgus posture whereas the unaffected leg displayed a neutral position in the left and right hemiplegic CP groups (Table 5).

## 4. Discussion

Most studies have focused on the characteristics of plantar pressure and gait postural analysis in children with CP. However, few studies currently have explored college and university students with CP, and even fewer studies have been conducted around the differences in foot types, plantar load associated with centers of gravity and foot posture between age-matched students of both sexes in the left and right hemiplegic CP conditions. Therefore, the present study was aimed not only at expanding the knowledge base regarding the characteristics of foot types and plantar pressure with the centers of gravity and the rearfoot posture in male and female students with CP during upright standing, but also at assessing the differences in these research items between the hemiplegic limbs and unaffected limbs in the left and right hemiplegic CP students. The plantar pressure measurement equipment used in this study has been applied in numerous previous research cases, including the foot and ankle research of special elite athletes [30,31,33] as well as a clinical study on common podiatric diseases [32]. Although the JC Mat differs from the Pedar-X^®^, which is considered the gold standard for in-shoe plantar pressure measurement [37], it has; however, been used both in clinical and commercial settings in Taiwan. In addition to providing information on plantar profiles and foot characteristics, the device used in the present study also allows for comparisons with the relevant research in the past.

The results of this study revealed that the values of the static bipedal AI of both feet in students with CP exhibited a significantly higher trend, particularly in the male students, who were categorized as the low arch. In the left and right hemiplegic CP groups, notably, the AI value was found to be significantly higher on their corresponding hemiplegic limbs. The result of the lower arch structure in CP college students was consistent with previous studies that revealed CP children have significantly higher AI values than normal children [6,38,39], suggesting that CP children bear a high probability of flat foot deformity of the foot. Changes in foot morphology are often thought to be closely related to differences in foot pressure distribution [38]. Static plantar pressure measurements can provide essential information regarding the loading of various parts of the foot and lower extremity, especially in patients with foot deformities and postural abnormalities caused by CP [6].

With respect to pressure distribution within the forefoot, midfoot and rearfoot regions, the static bipedal relative loads of the CP group were mainly focused on the left forefoot, and the dual regions of the forefoot and rearfoot of the right foot, particularly in the case of male CP students. Simultaneously, with reference to the results of the centers of gravity, it was shown that the CP group’s static centers of gravity tended to be on the right foot, particularly in male CP students (except the females). This may be due to a shift of weight to the right foot with dual plantar loading regions (forefoot and rearfoot) rather than the left foot with a single region (forefoot). With regards to the left hemiplegic condition, the relative loads on the left forefoot, and on the right forefoot and rearfoot were higher, thus, also revealing a similar phenomenon that corresponded to the findings of the shift of weight to the unaffected right foot with dual regions of relative loads. Similar results of left hemiplegia were found in the right hemiplegic condition. The relative load exerted more towards the forefoot region of both feet and also exhibited a shift of weight to the unaffected left foot.

The results were consistent with Femery et al.’s findings [18] that the propulsive foot (unaffected foot) had a 28% reduction in heel pressure peaks compared to the loading foot (affected foot). Further, the pressure peaks at the metatarsal heads and the hallux of the propulsive foot were relatively increased by 32%. Moreover, they noted that plantar pressure profiles differed significantly between unaffected and affected feet. According to their study, the affected heel pressure peaks in children with hemiplegic CP were decreased by approximately 21% and 97%, respectively. Galli et al. [6] and Righi et al. [38] investigated further, noting that the static contact area and plantar pressure of the forefoot were increased in the left and right hemiplegic CP children. Chang et al. [4] noted that plantar pressure mainly distributed at the first to fifth toes in children with mild CP while undergoing static assessments. The results could be explained by increasing anterior weight-bearing in CP children, who keep an antepulsion posture to maintain their abnormally balanced posture [38]. Nevertheless, there are some discrepancies between the results of this study and the aforementioned studies since they emphasized an increase in midfoot contact area and reduced rearfoot load as most of their subjects had flat feet. This may be attributed to the sample considered in this study, the differences may inevitably exist in age, gender and quantity of research participants, and measuring equipment. The other possible explanation was that children with spastic CP exhibited extensive stiffness on the ankle joint resulting in the shortening of their medial gastrocnemius and thus a restriction in their movement response ability [40]. Owing to their condition, CP patients must compensate by adopting an abnormal alignment in order to maintain body balance in a static position, such as keeping an antepulse posture [41]. This adaptation often requires considerable effort from the posterior antigravity muscles to avoid falls in daily life, which may lead to overloading of the posterior muscle chain [42]. Children with CP can engage in motor postural adjustment activities apart from these compensatory strategies, but their lower extremity muscles are typically activated to a lesser extent and with a delayed muscular transmission compared to healthy individuals [43,44].

As for the results of centers of gravity transferred toward the unaffected limb in hemiplegic CP students, the situation can be referred to previous studies which indicated hemiplegia to be characterized by dyskinesia and unilateral spasticity of the contralateral upper and lower extremities of the affected cerebral hemisphere, resulting in a characteristic compensatory use of the unaffected half of the body, thereby impeding weight transfer on the affected side [41,42]. Furthermore, examination of the leg-length discrepancy of the limb, which was not included in this study, is likely to bear an impact on changes of plantar pressure associated with the balance of the centers of gravity in CP patients. As shown in the studies by Pereiro-Buceta et al. [45], the increased discrepancy of leg-length resulted in asymmetric foot-loading with an increased mean and peak pressure on the short limb and decreased plantar loads on the long limb. A greater discrepancy may lead to an increased stance time on the long limb. Nevertheless, the discrepancy in limb-length can be associated with further gait deviations [46]. This study did not include the investigation concerning the leg-length discrepancy. Given the potential impact of the discrepancy on the limbs and plantar pressures, it is acknowledged that the discrepancy which occurs between the unaffected and hemiplegic limb in CP patients and the dynamic plantar profiles during the walking phase warrants further studies.

With respect to the results of the detailed six subregional plantar pressure distribution, the relative load was commonly exerted beneath the medial metatarsals of both feet in CP students. In addition to the medial metatarsal, the male CP students’ relative load also mainly distributed to the entire heels of both feet, while the relative load of female CP students was significantly increased at the medial longitudinal arch of both feet.

This study seemed to parallel the studies by Galli et al. [6], which indicated that plantar–flexor hypermobility and knee flexion, as well as a tendency toward flat arches in CP children lead to an increased plantar load of the forefoot. Furthermore, the results of the present study were consistent with previous studies which maintained that pressure peaks at the metatarsal heads and the hallux were increased beneath the propulsive foot [18]. The hallux, in particular, showed an insufficient push-off on the affected side. The ratio of pressure abnormalities was commonly observed at the hallux and the first metatarsal of CP children [19].

As for the results of plantar pressure distribution in hemiplegic CP condition, we discovered that the common feature in plantar load was mainly exerted on the medial metatarsal of both feet in the left and right hemiplegic CP students. However, it is worth noting that a traceable feature was found, and this reflected an increased plantar load displayed on the heel of the unaffected limb. Furthermore, plantar loads were considerably concentrated more on the medial aspect of the hemiplegic foot. To some extent, the findings echoed the evidence that children with hemiplegic CP suffered from reduced pressure peaks under the affected heel [18]. However, in comparison with the results of the static rearfoot postural alignment, findings from this research showed that the hemiplegic foot presented a rearfoot valgus posture in both hemiplegic groups. Corresponding with the studies by Leunkeu et al. [19], findings from this research revealed that the plantar pressure peaks were distributed at the hallux and metatarsal head of the foot. Increased contact with the forefoot region may represent a compensatory strategy to increase standing and walking stability and improve low limb support for CP patients [47]. Moreover, pes planovalgus (flat feet) has been recognized as a common feature of foot deformity in children with CP [48]. Ehlert et al. [49] investigated further, asserting that the flat feet characteristics with pronation of the calcaneus are often observed in CP children. Apart from this, the hemiplegic CP students in this study presented a most common feature of increased plantar load on the medial aspect of the foot, accompanied by a rearfoot valgus posture on the hemiplegic foot. The feature appeared to constitute the pronated foot characteristics, and such results were consistent with previous studies that suggested bearing a lower medial longitudinal arch contributes to the incidence of rearfoot pronation and thus increased plantar load on the medial aspect of the rearfoot [32,50,51]. Rearfoot valgus associated with a pronated foot often significantly increases the possibility of extensive foot pain [52]. Furthermore, Stovitz and Coetzee’s study noted that the forefoot abduction and increased pressure on the thumb are susceptible to complications of the first metatarsal syndrome (e.g., hallux valgus) and the second metatarsal syndrome (e.g., metatarsalgia) [53]. Since the first and second metatarsal syndromes are caused by an increased hallux plantar loading which may later develop into functional hallux limitus [54], thus, the limited dorsiflexion hallux and first metatarsophalangeal joint range of motion that occur on the hemiplegic foot of patients with CP may be an important traceable feature that should be further explored and referred to the measurement method proposed by Sánchez-Gómez et al. [54] in future studies.

This study functioned as one of the pioneers of investigating the characteristics of foot types and plantar pressure associated with the centers of gravity, and the rearfoot posture among male and female college students in the left and right hemiplegic CP condition, respectively. However, some limitations which may exist in the study were: first, the sample size was small, and the participants were college and university students in Taiwan; thus, limiting the generalizability and commonality of the findings in this study. Second, since the participants in this study were non-randomly sampled, the sampling method may lead to an unavoidable systematic error. Considering the participants were not fully representative, sampling bias may affect the interpretation of the results. Third, each participant in this study wore an AFO for different seniority, and this may be one of the factors which affected the results to a certain extent. However, it is worth noting that the results of this study could reveal the characteristics of plantar pressure profiles and foot postures among male and female students with left and right hemiplegic CP. In addition, our findings also revealed a traceable feature for the foot pressure and posture between the hemiplegic limb and unaffected limb. These results may prove beneficial for further providing a strategy in designing a customized insole and orthosis with considerations of accurate height, wedge and angle for adjusting the uneven distribution of plantar pressure to improve footwear comfort and fall prevention in CP patients.

## 5. Conclusions

We concluded that college and university students with CP exhibited a low-arched foot, particularly on their corresponding hemiplegic limbs. Static foot pressures and centers of gravity indicated that weight was shifted to the unaffected foot with dual plantar loading regions (forefoot and rearfoot) rather than the hemiplegic foot with a single loading region (forefoot). In addition, the relative load was commonly exerted at the medial metatarsals of both feet in CP students. Moreover, the male CP students’ relative load mainly distributed to the entire heels of both feet, while the female CP students’ relative load was significantly increased at the medial longitudinal arch of both feet. A common feature of both the left and right hemiplegic CP conditions was an increase in relative load at the medial metatarsal of both feet. A traceable feature from these findings is that there was an increased relative load on the heel of the unaffected limb, and the load was also relatively concentrated on the medial aspect of the hemiplegic foot. The pressure profiles resonated with the rearfoot valgus posture of the hemiplegic foot, and this could serve as a traceable beginning for possible links among the pronated low-arched feet, mild centers of gravity, metatarsal syndrome and rearfoot valgus of hemiplegic limbs in CP patients.

## Figures and Tables

**Figure 1 jpm-12-00394-f001:**
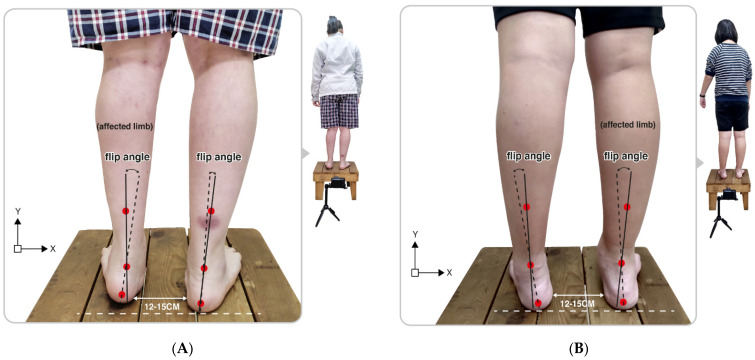
Illustration of the upright posture of the representative participant in (**A**) the left hemiplegic CP and (**B**) the right hemiplegic CP group. Flip angle (static rearfoot angle) was calculated based on the intersection of two straight lines produced from the three positioning markers on the lower legs.

**Figure 2 jpm-12-00394-f002:**
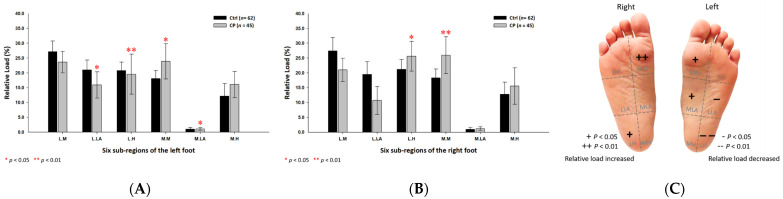
The percentage of six subregional relative loads of (**A**) left and (**B**) right feet in total students during static stance. The changes in pressure distribution are illustrated in (**C**) foot plantar diagram. Significant differences between the control group and the CP group were determined as * *p* < 0.05 and ** *p* < 0.01. Six subregions of the foot and the abbreviations are as follows: LM lateral metatarsal bone; LLA, lateral longitudinal arch; LH, lateral heel; MM, medial metatarsal bone; MLA, medial longitudinal arch and MH, medial heel.

**Figure 3 jpm-12-00394-f003:**
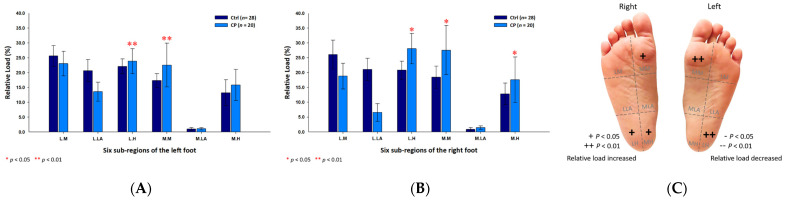
The percentage of six subregional relative loads of (**A**) left and (**B**) right feet in male students during static stance. The changes in pressure distribution are illustrated in (**C**) foot plantar diagram. Significant differences between the male control group and the male CP group were determined as * *p* < 0.05 and ** *p* < 0.01.

**Figure 4 jpm-12-00394-f004:**
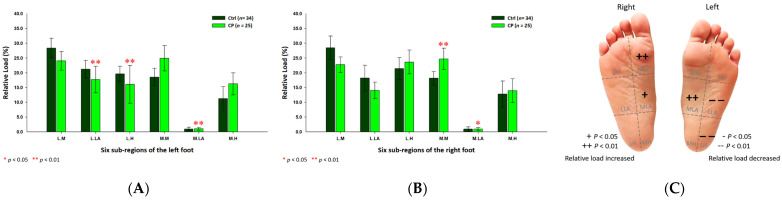
The percentage of six subregional relative loads of (**A**) left and (**B**) right feet in female students during static stance. The changes in pressure distribution are illustrated in (**C**) foot plantar diagram. Significant differences between the female control group and the female CP group were determined as * *p* < 0.05 and ** *p* < 0.01.

**Figure 5 jpm-12-00394-f005:**
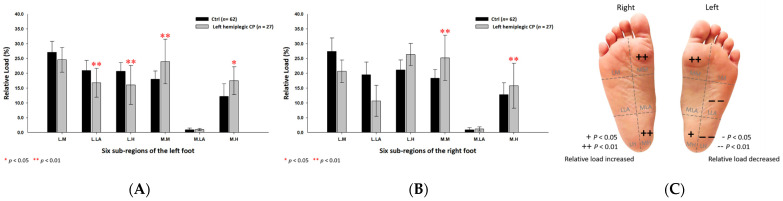
The percentage of six subregional relative loads of (**A**) left and (**B**) right feet in total students during static stance. The changes in pressure distribution are illustrated in (**C**) foot plantar diagram. Significant differences between the control group and the left hemiplegic CP group were determined as * *p* < 0.05 and ** *p* < 0.01.

**Figure 6 jpm-12-00394-f006:**
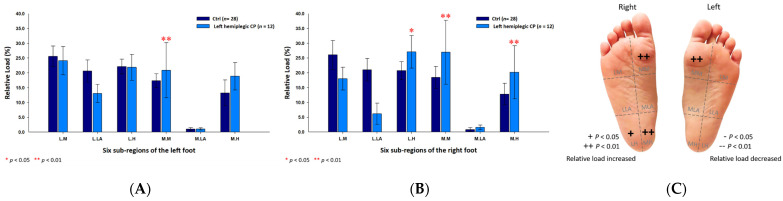
The percentage of six subregional relative loads of (**A**) left and (**B**) right feet in male students during static stance. The changes in pressure distribution are illustrated in (**C**) foot plantar diagram. Significant differences between the male control group and the male left hemiplegic CP group were determined as * *p* < 0.05 and ** *p* < 0.01.

**Figure 7 jpm-12-00394-f007:**
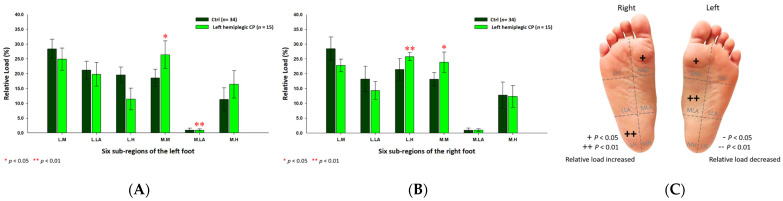
The percentage of six subregional relative loads of (**A**) left and (**B**) right feet in female students during static stance. The changes in pressure distribution are illustrated in (**C**) foot plantar diagram. Significant differences between the female control group and the female left hemiplegic CP group were determined as * *p* < 0.05 and ** *p* < 0.01.

**Figure 8 jpm-12-00394-f008:**
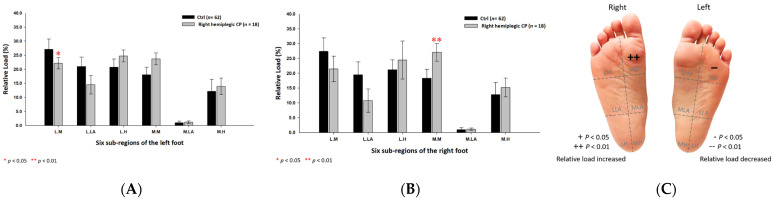
The percentage of six subregional relative loads of (**A**) left and (**B**) right feet in total students during static stance. The changes in pressure distribution are illustrated in (**C**) foot plantar diagram. Significant differences between the control group and the right hemiplegic CP group were determined as * *p* < 0.05 and ** *p* < 0.01.

**Figure 9 jpm-12-00394-f009:**
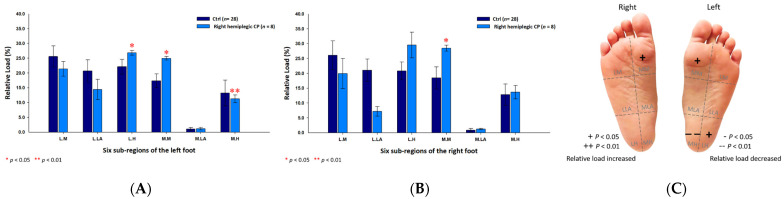
The percentage of six subregional relative loads of (**A**) left and (**B**) right feet in male students during static stance. The changes in pressure distribution are illustrated in (**C**) foot plantar diagram. Significant differences between the male control group and the male right hemiplegic CP group were determined as * *p* < 0.05 and ** *p* < 0.01.

**Figure 10 jpm-12-00394-f010:**
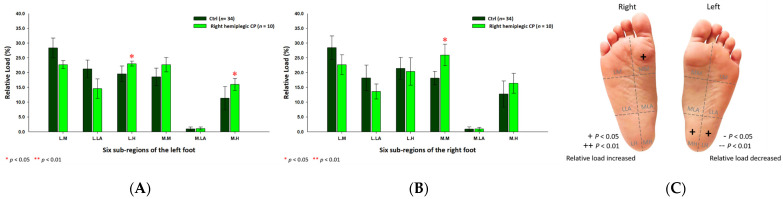
The percentage of six subregional relative loads of (**A**) left and (**B**) right feet in female students during static stance. The changes in pressure distribution are illustrated in (**C**) foot plantar diagram. Significant differences between the female control group and the female right hemiplegic CP group were determined as * *p* < 0.05 and ** *p* < 0.01.

**Figure 11 jpm-12-00394-f011:**
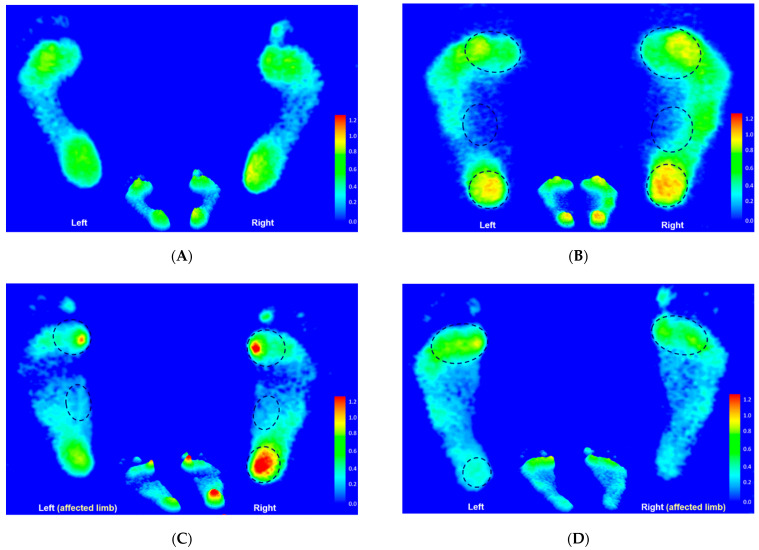
The static footprint characteristics of the representative participant in the control group (**A**), the CP group (**B**), the left hemiplegic CP group (**C**), and the right hemiplegic CP group (**D**). Black circles indicate the areas of higher relative loading.

**Table 1 jpm-12-00394-t001:** Summary of the demographic characteristics for students diagnosed with hemiplegic cerebral palsy and healthy college students.

	Healthy			Hemiplegic Cerebral Palsy		
	Total	Male	Female	Total	Male	Female
Number	62	28	34	45	20	25
Age (years)	20.2 ± 0.8	20.1 ± 0.9	20.2 ± 0.7	19.4 ± 0.9	18.8 ± 0.5	20.0 ± 0.7
Height (cm)	165.5 ± 5.5	168.2 ± 5.1	163.3 ± 4.8	158.4 ± 4.8	161.4 ± 4.3	156.0 ± 3.8
Mass (kg)	60.8 ± 4.1	63.6 ± 3.8	58.5 ± 2.5	53.5 ± 4.7	57.0 ± 3.8	50.8 ± 3.5
BMI (kg/m^2^)	22.2 ± 1.2	22.5 ± 1.1	22.0 ± 1.3	21.4 ± 1.8	21.9 ± 1.6	20.9 ± 1.8

Abbreviations: BMI, body mass index. Data are presented as mean ± standard deviation (SD).

**Table 2 jpm-12-00394-t002:** Static arch index of both feet for students diagnosed with hemiplegic cerebral palsy and healthy college students.

	Left Foot			Right Foot		
	Total	Male	Female	Total	Male	Female
Healthy	0.22 ± 0.04	0.24 ± 0.02	0.20 ± 0.05	0.22 ± 0.05	0.23 ± 0.03	0.21 ± 0.06
CP group	0.26 ± 0.03	0.27 ± 0.03 ^1^	0.24 ± 0.03 ^1^	0.26 ± 0.04	0.27 ± 0.05 ^1^	0.25 ± 0.03 ^1^
Left hemiplegic CP	0.26 ± 0.03	0.27 ± 0.03 ^1^	0.25 ± 0.02 ^1^	0.26 ± 0.04	0.26 ± 0.04	0.26 ± 0.03
Right hemiplegic CP	0.25 ± 0.04	0.26 ± 0.05	0.24 ± 0.03	0.26 ± 0.04	0.27 ± 0.06 ^2^	0.24 ± 0.02 ^1^

Data are presented as mean ± standard deviation (SD). Significant differences were noted as ^1^ *p* < 0.05 and ^2^ *p* < 0.01, and determined by the independent sample *t*-test between the control group (n = 62; male = 28, female = 34) and the CP group (n = 45; male = 20, female = 25).

**Table 3 jpm-12-00394-t003:** Percentage of static relative load of the forefoot, midfoot and rearfoot regions for students diagnosed with hemiplegic cerebral palsy and healthy college students.

	Left Foot (%)			Right Foot (%)		
	Total	Male	Female	Total	Male	Female
Forefoot						
Healthy	17.74 ± 11.40	21.48 ± 5.10	14.65 ± 14.00	22.84 ± 5.95	22.27 ± 5.76	23.30 ± 6.11
CP group	23.73 ± 4.89 ^2^	22.79 ± 5.93	24.47 ± 3.77 ^2^	23.48 ± 5.77 ^1^	23.18 ± 7.90 ^1^	23.72 ± 3.27 ^2^
Left hemiplegic CP	24.28 ± 6.02 ^2^	22.54 ± 7.43	25.67 ± 4.23 ^2^	22.95 ± 6.40	22.49 ± 9.14 ^1^	23.32 ± 2.87 ^2^
Right hemiplegic CP	22.90 ± 2.20 ^2^	23.18 ± 2.55 ^2^	22.68 ± 1.92 ^2^	24.27 ± 4.64 ^1^	24.21 ± 5.66	24.31 ± 3.80 ^2^
Midfoot						
Healthy	10.98 ± 10.31	10.85 ± 10.26	11.10 ± 10.43	10.19 ± 9.84	10.93 ± 10.55	9.58 ± 9.24
CP group	8.48 ± 8.08 ^2^	7.36 ± 6.72 ^2^	9.38 ± 8.99 ^2^	5.95 ± 5.85 ^2^	4.01 ± 3.34 ^2^	7.51 ± 6.90 ^2^
Left hemiplegic CP	8.92 ± 8.69 ^2^	7.08 ± 6.50 ^2^	10.40 ± 9.97	5.97 ± 6.05 ^2^	3.87 ± 3.45 ^2^	7.64 ± 7.14 ^2^
Right hemiplegic CP	7.82 ± 7.16 ^2^	7.78 ± 7.24 ^2^	7.85 ± 7.28 ^2^	5.93 ± 5.61 ^2^	4.21 ± 3.27 ^2^	7.30 ± 6.72 ^2^
Rearfoot						
Healthy	16.46 ± 5.61	17.68 ± 5.72	15.45 ± 5.35	16.97 ± 5.60	16.80 ± 5.20	17.12 ± 5.95
CP group	17.80 ± 5.95	19.85 ± 6.22	16.17 ± 5.22	20.57 ± 7.52 ^2^	22.82 ± 8.35 ^2^	18.77 ± 6.32
Left hemiplegic CP	16.80 ± 5.72	20.39 ± 4.68	13.94 ± 4.82	21.08 ± 7.97 ^2^	23.63 ± 8.09 ^2^	19.03 ± 7.38 ^2^
Right hemiplegic CP	19.30 ± 6.04	19.04 ± 8.11	19.52 ± 3.90	19.81 ± 6.84	21.59 ± 8.84	18.38 ± 4.43

Data are presented as mean ± standard deviation (SD). Significant differences were noted as ^1^ *p* < 0.05 and ^2^ *p* < 0.01, and determined by the independent sample *t*-test between the control group (n = 62; male = 28, female = 34) and the CP group (n = 45; male = 20, female = 25).

**Table 4 jpm-12-00394-t004:** Percentage of centers of gravity in static standing for students diagnosed with hemiplegic cerebral palsy and healthy college students.

	Left Foot (%)			Right Foot (%)		
	Total	Male	Female	Total	Male	Female
Control group	48.85 ± 3.70	49.39 ± 3.21	48.41 ± 4.05	51.16 ± 3.69	50.61 ± 3.21	51.62 ± 4.03
CP group	48.60 ± 7.96 ^2^	48.55 ± 8.08 ^2^	48.64 ± 8.04	51.40 ± 7.96 ^2^	51.45 ± 8.08 ^2^	51.36 ± 8.04 ^2^
Left hemiplegic CP	43.41 ± 5.22 ^1^	42.83 ± 3.90 ^2^	43.87 ± 6.17 ^1^	56.59 ± 5.22 ^2^	57.17 ± 3.90 ^2^	56.13 ± 6.17 ^2^
Right hemiplegic CP	56.39 ± 3.93 ^2^	57.13 ± 3.60 ^2^	55.80 ± 4.26 ^2^	43.61 ± 3.93 ^2^	42.88 ± 3.60 ^2^	44.20 ± 4.26 ^1^

Data are presented as mean ± standard deviation (SD). Significant differences were noted as ^1^ *p* < 0.05 and ^2^ *p* < 0.01, and determined by the independent sample *t*-test between the control group (n = 62; male = 28, female = 34) and the CP group (n = 45; male = 20, female = 25).

**Table 5 jpm-12-00394-t005:** Static rearfoot postural alignment for students diagnosed with hemiplegic cerebral palsy and healthy college students.

	Left Foot			Right Foot		
	Total	Male	Female	Total	Male	Female
Healthy	4.10 ± 2.41	4.04 ± 2.38	4.15 ± 2.47	4.68 ± 1.81	4.76 ± 1.55	4.61 ± 2.01
CP group	5.18 ± 1.59	5.12 ± 1.42	5.23 ± 1.74	4.48 ± 1.46	4.30 ± 1.09	4.62 ± 1.71
Left hemiplegic CP	6.11 ± 1.25 ^1^	6.15 ± 0.67 ^1^	6.07 ± 1.60	3.69 ± 1.32 ^1^	3.51 ± 0.52 ^1^	3.84 ± 1.72
Right hemiplegic CP	3.80 ± 0.87 ^2^	3.57 ± 0.47 ^1^	3.98 ± 1.09 ^1^	5.65 ± 0.66 ^2^	5.48 ± 0.36 ^1^	5.79 ± 0.82 ^1^

Data are presented as mean ± standard deviation (SD). Significant differences were noted as ^1^ *p* < 0.05 and ^2^ *p* < 0.01, and determined by the independent sample *t*-test between the control group (n = 62; male = 28, female = 34) and the CP group (n = 45; male = 20, female = 25).

## Data Availability

The datasets generated and/or analyzed for the present study are available from the corresponding author upon reasonable request.

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
