# Peer review of "Traceable Features of Static Plantar Pressure Characteristics and Foot Postures in College Students with Hemiplegic Cerebral Palsy"

_jpm, 2022, doi:10.3390/jpm12030394_

Round 1

Reviewer 1 Report

My comments are predominantly  for the content, but do have some edits/suggestions to improve readability:

1) Title: The title is correct as it reflects correctly the objective and hypothesis of the work.

2) Summary: This section follow a well structured format.

3) Introduction: The research question itself is sound and the topic is strongly introduced. On the other hand, Introduction section may be improved adding new information in order to provide an adequate state-of-the-art including some references.

I suggest to include to following reference to complete the requerimient:

 doi.org/10.5114/aoms.2020.97053

4) Materials and Methods: The inclusion and exclusion criteria are adequate. Sampling bias needs to be discussed within the limitations section given the study design.

5) Results: The results is clear and concise with appropriate statistical analysis been performed appropriately and rigorously.

6) Discussion: The discussion appears well developed and appropriate, authors describe the results, the limitations and compare with other researchs. Hower author should compare their results with others similar researches related to biomechanics activities. I suggest to include the following reference to complete the requeriment:

 doi: 10.1177/1071100719901116

doi: 10.3390/healthcare9080932.

The limitations section needs to incorporate the sampling bias given the study design.

7) Conclusion: The conclusion is conclusively.

9) Figures and tables: Correct

Author Response

Dear Reviewer,

Thank you for the constructive suggestions and comments on my manuscript (ID: jpm-1597845). The suggestions and comments are helpful for improving the manuscript. I am submitting the revised version of the manuscript with my responses to the suggestions and comments by the reviewer. Many thanks for your guidance.

My responses to each suggestion and comment are as follows, and they are presented in blue texts with a grey background color in the revised manuscript.

Reviewer 2 Report

1. Overall Strengths
The manuscript has not have reach merits. The number of participants is not significance, very short with 45 participants, and poor description of the methodology/statistics. The study uses a not gold estándar tool for evaluate the foot type and plantar pressures the pedigraph image of static dominant footprint was taken to assess foot type is not a validate tool, in general to be poor planned. The text is structured and is difficult to follow. there are several language related errors and thus the manuscript requires a thorough checking by a native speaker of English.
2. Importance
The topic has not potential to offer more solid evidence based information about the potential of clinical outcome of plantar pressure and traceable Features of Static Plantar Pressure Characteristics and Foot Postures in College Students with Hemiplegic Cerebral Palsy
Hypotheses are needed. What do the authors expect to find based on previous work? This would also be a good place to justify why this study is needed.

3. Justification/Rationale
The justification of this study could be strengthened by explaining what this study adds to the pre-existing literature. One very important finding of the study should appear already in Abstract or in Conclusions: Even though the findings showed a traceable feature to a possible connection among the pronated low arches, mild centers of gravity, metatarsal syndrome and rearfoot valgus of the hemiplegic limbs in CP patients.
Please make it clearer already in the Introduction, what new this study has to offer (i.e. in what way it differs from earlier studies).
4. Methods/Approach
The methodology is not rationale, seriously flawed from the methodological point of view related type of the design and case control study; including key details such as how many patients they need to recruit, what their outcomes will be, how they intend to measure these outcomes, etc.
The tool used used are not standardized and not well suited for the purpose. In first place, the plantar pressure was measured using the 
optical plantar pressure sensitive mat (JC Mat, View Grand International Co Ltd, New Taipei City, Taiwan), but pressure platform is not validated previously, even to assess plantar pressures or any other medical condition. If there´s not validation, the values find in this study are not reliable. In second place, pedigraph image of static dominant footprint was taken to assess foot type, this tool is not a gold standar for evaluate this condition and not well suited for the purpose. I appreciate that the authors included a sample size calculation
How was assessed the calibration of the platform ? A external process of calibration is needed in a high standard level of pressures measurements. If the authors wants to “validate” this clinical-marketing tool for a research propose. This pressure platform don’t have this process, so it´s automatically discarded to a good research propose.

5. Results/Findings
These issues invalidates completely all the trial

6. Discussion
Discussion is muddled, confusing to follow and repeats somewhat the Introduction. Furthermore, some more shortcomings should be included, particularly the fact the bias found. The authors could also consider some more ideas for further studies.

7. Conclusions
Write this section part again and clearly.

8. Figures.
please reconsider listing of the all figures and remove this figures are not informative.

Author Response

(The authors gave the same response as above.)

Reviewer 3 Report

The article addresses an extremely relevant topic. Overall the article is well structured. The introduction is complete and well written. The methods well adapted to the objectives, as well as the discussion and conclusions.
However, minor points must be improved.
Regarding tables, legends should be placed at the top of the table. under the table only statistical significance should be listed.
In subheadings 2.3; 3.2; 3.5; 3.6 Sentences must not begin with an abbreviation.

Author Response

(The authors gave the same response as above.)

Round 2

Reviewer 1 Report

Autors have adressed all my requeriment in the correct way

Author Response

Dear Reviewer,

I sincerely appreciate for your constructive suggestions and comments on my resubmitted manuscript (ID: jpm-1597845). The suggestions and comments are helpful for improving my manuscript. Many thanks for your guidance.

Thank you very much for your consideration of this manuscript.

Sincerely yours,

Tong-Hsien Chow, Ph.D.

Reviewer 2 Report

My opinion about the article remains the same of the first revision of manuscript. Most of the issues that I advanced to you cannot be repaired. A new study would be needed to make these things suitable. The clarifications provided do not solve the problem. 

Author Response

(The authors gave the same response as above.)
